# The Effect of Temperature on the Growth of Holopelagic *Sargassum* Species

Edén Magaña-Gallegos [1], Eva Villegas-Muñoz [2], Evelyn Raquel Salas-Acosta [1], M. Guadalupe Barba-Santos [1], Rodolfo Silva [3] and Brigitta I. van Tussenbroek [1,*]

[1] Unidad Académica de Sistemas Arrecifales, Instituto de Ciencias del Mar y Limnología-UNAM, Prol. Av. Niños Héroes S/N, Puerto Morelos 77580, Mexico

[2] Facultad de Estudios Superiores Iztacala, Universidad Nacional Autónoma de México, Los Reyes Iztacala, Av. de los Barrios # 1, Mexico City 54090, Mexico

[3] Instituto de Ingeniería, Universidad Nacional Autónoma de México, Mexico City 04510, Mexico

[*] Correspondence: vantuss@cmarl.unam.mx

**Abstract:** Holopelagic *Sargassum* species have bloomed recurrently in the northern tropical Atlantic since 2011, causing socioeconomic and environmental problems. Little is known about their basic biology and responses to the abiotic environment. The aim of this study was to determine how temperature affects the growth rates of the genotypes *S. fluitans* III, *S. natans* I, and *S. natans* VIII that predominate in these blooms. The growth rates were evaluated in specially designed ex situ systems between 22 and 31 °C, which corresponds with the natural temperature range of these seaweeds in the northern tropical Atlantic. All the genotypes had decreased growth rates at 31 °C, and they varied in their response to temperature, with *S. fluitans* III presenting a maximal rate of 0.096 doublings· day$^{-1}$ (doubling its weight in 10.5 d) at 28 °C and *S. natans* VIII a minimal rate of 0.045 doublings· day$^{-1}$ (doubling its weight in 22.2 d) at 31 °C. In addition, the response to the temperature varied depending on the time of the year. Understanding the role of temperature in the growth of holopelagic *Sargassum* genotypes, amongst other factors influencing their physiology (such as nutrients, salinity tolerance, or light, including their interactions), could help to understand the dynamics of the recent blooms in the tropical North Atlantic.

**Keywords:** sargasso; algal bloom; temperature range; specific growth rate; doubling time; season

## 1. Introduction

Since 2011, recurrent blooms of holopelagic *Sargassum* species (referred to as sargasso herein) have occurred in the northern tropical Atlantic [1], forming the "Great Atlantic Sargassum Belt", that extends from the coasts of tropical Africa to the Gulf of Mexico [1,2]. The algae responsible for these blooms are *S. fluitans* (morphotype III) and *S. natans* (morphotypes I and VIII), based on the phenotypic traits, and more recently, on genetic characteristics [3–5]; herein, we refer to the species and their morphotypes as genotypes. In the past, sargasso was only abundant in the Sargasso Sea, with the predominance of *S. natans* I and *S. fluitans* III [3]. The previously rare *S. natans* VIII predominated in early post-2011 blooms events in the tropical North Atlantic [3]. However, its relative abundance has decreased in recent years, and the predominance of the genotypes vary both intra-annually and seasonally, with *S. fluitans* III gradually increasing in overall predominance [6].

Sargasso travels ~ 8850 kms from West Africa to the Caribbean and the Gulf of Mexico [1], and throughout this journey, the algae cope with numerous physical and chemical drivers, which probably affect the growth and physiology [1,2]. The salinity, irradiance, nutrient availability (e.g., nitrate and phosphate), and temperature depend on regional (e.g., terrestrial river runoffs and upwellings) and large-scale oceanic and meteorological processes in the North Atlantic (the North Atlantic Oscillation, the Atlantic Niño, the Atlantic Meridional Mode, etc.), which are likely associated with sargasso blooms [1,2,7].

The effect of the seawater temperature on the growth of sargasso is the focus of this study, as previous studies have indicated that temperature is important for the phenology of seaweeds, including benthic *Sargassum* species [8–13]. However, to our knowledge, only one study has systematically related temperature to growth rates in experimental series [14]. However, the sargasso in the study of Hanisak and Samuel [14] was of Sargasso Sea origin, and the employed resolution of temperatures, at increments of 6 °C from 18 °C to 30 °C, made it difficult to determine the optimal temperature for growth. Others have studied the growth of sargasso for various purposes at different ambient temperatures (e.g., [15–17]), but their designs did not allow for drawing conclusions concerning the growth response of these algae to temperature.

Inferences concerning the significance of the seawater temperature on the recent blooms suggest that sargasso grows faster in the warmer waters of the Great Atlantic Sargassum Belt (GASB) than the generally cooler Sargasso Sea. However, within the GASB, sargasso seems to grow better at lower temperatures, although colder upwelling waters, rich in nutrients, may be responsible for the higher growth rates [1,2,18]. The average ocean sea surface temperatures were lower in years with exceptional blooms [1], and a negative phase of the Atlantic meridional mode causing cooling and movement of the Intertropical Convergence Zone was linked to the more intense blooms of 2015 and 2018 [2]. The purpose of this research is to determine how temperature affects the growth rates of three genotypes of sargasso, in specially designed ex situ systems that keep sargasso in motion, thereby maintaining the algae in good condition [17]. We hypothesize that the different genotypes respond differently to changing temperatures (considering the interannual and seasonal differences in their relative dominance), and that the highest temperatures inhibit the growth of these algae.

## 2. Materials and Methods

### 2.1. Sargasso Collection

Drifting sargasso was collected by boat or kayak in the Puerto Morelos reef lagoon (20°52' N, 86°52' W, >70 m from the shoreline) and classified by genotype (*S. fluitans* III, *S. natans* I, and *S. natans* VIII), using previously described criteria [3–6]. Sargasso was collected using nets and examined for characteristics such as an attractive (light brown) color, having no breaks or symptoms of decay. The collected specimens were placed in portable hard coolers, filled with seawater, and transported within 20 min to the experimental location at the Reef Systems Unit, Universidad Nacional Autónoma de México (UNAM). The sargasso was brushed clean to eliminate any debris and epibionts before being placed in the experimental area. Because the hydroids were firmly attached to the blades, they were left in place. Small apical sections weighing 6 g wet weight were separated and used for the experiments.

### 2.2. Description of the Ex Situ Culture System

Specially designed closed continuous motion systems [17] were used to observe the growth rates of the sargasso (Figure 1). Each system included a 52 × 35 × 25 cm (approx. 45 L) sump, from which the water was pumped up to an 18 L bucket. To achieve the appropriate temperature, the sumps of the four systems were immersed in chilled seawater kept in a fiberglass tank (Figure 1). The setup included a total of 16 buckets in four fiberglass tanks. The tank water was never mixed with the sump water. The water temperatures were maintained within 1.0 °C of the target temperature, using chilling and heating systems. The water was taken from the mesotrophic Puerto Morelos reef lagoon (salinity 35 and pH 8.2) via a >1 km long PVC tube and passed through a sand filter.

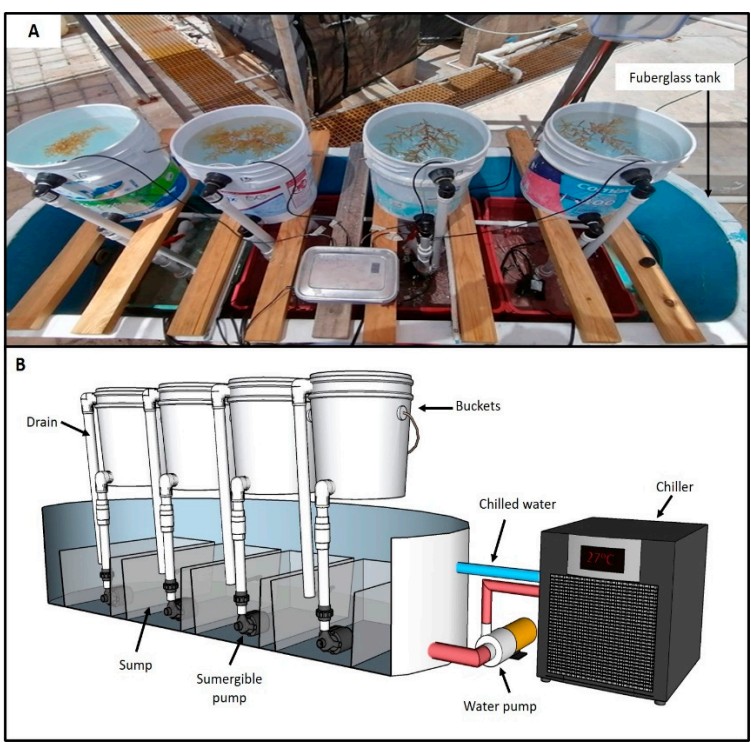

**Figure 1.** Ex situ culture system for sargasso. (**A**) Culture systems with sargasso and the (**B**) main components of the system.

### 2.3. Experimental Design

From 14 April to 1 November 2021, three experimental series with trials lasting five days were conducted to contrast the growth rates between the genotypes of sargasso. The experimental series were: (1) *S. fluitans* III vs. *S. natans* VIII, (2) *S. fluitans* III vs. *S. natans* I, and (3) *S. natans* I vs. *S. natans* VIII, at temperatures of at 22, 25, 28, and 31 °C. Because there were only two buckets per genotype for each temperature, several trials per experimental series were conducted over time. The collected algae were acclimatized in the ex situ culture system for one full day before the trials, at the prevailing ambient temperatures at the times of collection, which varied between 26.9 and 30.1 °C. The first experimental series was conducted from 14 April to 27 May (spring) and consisted of four trials (n = 7 per genotype × temperature), the second from 2 July to 20 July (summer) and consisted of two trials (n = 4 per genotype × temperature), and the third from 2 October to 1 November (autumn) and consisted of three trials (n = 6 per genotype × temperature). The number of trials was limited by COVID-19 restrictions and the variable availability of genotypes [6]. For the treatments at 22 °C and 25 °C, the heat pumps Delta Star DSHP-9 and Delta Star DSHP-7 (Aqualogic, San Diego, CA, USA) were used, respectively. For the treatment at 28 °C, a heat pump Inter Heat Plus 13P (Inter Heat, Jiangsu, China was used. For the 31 °C treatment, only heaters were used. The daily rate of the water exchange was 10%. The light was between 435 and 581 µmol m$^{-2}$ s$^{-1}$ (LI-1500, LI-COR, NE, USA), exceeding the determined saturation irradiance in natural conditions for sargasso, which was 200 to 300 µmol m$^{-2}$ s$^{-1}$ [18]. The starting and final wet weight measurements were used to calculate the growth rates, and the relative growth rate (RGR) was estimated as doubling per day following [14,17]. The detached tissue on the bottom of the buckets was collected, which consisted of the shed sargasso fragments that lost buoyancy. The growth was also measured by the increase in the number of internodes. A colored ribbon was tied to the stipe, five nodes from the apex of a branch, and four branches were marked with different colored ribbons per thallus. For this measurement, the thalli were allowed to grow for 10 days instead of 5, because the increase in the number of nodes was not always obvious after only five days.

*2.4. Statistical Analysis*

The normality and homogeneity of the variance of each set of data were examined. To compare the treatments, a factorial ANOVA was used, with the genotype and temperature as factors, along with their interactions. A factorial ANOVA analysis was also performed for each genotype, with the experimental series (realized in different seasons) and temperature as factors, as well as their interactions; this study was performed after we observed, against our expectations, that the growth rates of *S. natans* VIII differed between the experimental series. The variable analyzed was the relative growth rate (RGR). Tukey's test was used whenever there were significant differences [19]. All tests were run with a significance level of 5%. Statistical analyses were carried out using the RStudio program version 4.2.2 [20].

## 3. Results

The three genotypes had high growth rates throughout the tested temperature range, with an RGR between 0.045 and 0.095 doublings $\cdot$ d$^{-1}$, which corresponded with 22.2 and 10.5 days to double their weight, respectively (Table 1). The growth rates differed among genotypes and temperatures, and the general tendencies in rates, either determined as the RGR or an increment in number of nodes, coincided (Table 1). The RGR proved to be more precise to measure the growth; thus, we used this measure for further discussion. The weight of the detached tissues (0.12 wet g on average) was small in comparison with the 6 g wet weight of the thalli used for the experiments, and the weight of the detached tissues did not differ among genotypes or temperature treatments (Table 1).

**Table 1.** The means ($\pm$SE) of the relative growth rates, the weight of the detached fragments, and the increase in the internodes of the holopelagic *Sargassum* genotypes at different temperatures. The time to double weight is derived from the mean RGR (1/mean RGR). Different superscript letters indicate significant differences (one-way ANOVA, post-hoc Tukey test, $p < 0.05$).

| Temperature | 22 °C | 25 °C | 28 °C | 31 °C |
|---|---|---|---|---|
| **Relative growth rate (doubling $\cdot$ d$^{-1}$)** | | | | |
| *S. fluitans* III | 0.078 [ab] $\pm$ 0.01 | 0.077 [ab] $\pm$ 0.1 | 0.095 [a] $\pm$ 0.01 | 0.058 [b] $\pm$ 0.01 |
| *S. natans* I | 0.057 $\pm$ 0.01 | 0.067 $\pm$ 0.01 | 0.063 $\pm$ 0.01 | 0.054 $\pm$ 0.01 |
| *S. natans* VIII | 0.058 $\pm$ 0.01 | 0.059 $\pm$ 0.01 | 0.053 $\pm$ 0.01 | 0.045 $\pm$ 0.01 |
| **Time to double weight (d)** | | | | |
| *S. fluitans* III | 12.8 | 13.0 | 10.5 | 17.2 |
| *S. natans* I | 17.5 | 14.9 | 15.9 | 18.5 |
| *S. natans* VIII | 17.2 | 16.9 | 18.9 | 22.2 |
| **Detached fragments (wet mg)** | | | | |
| *S. fluitans* III | 124 $\pm$ 48 | 125 $\pm$ 48 | 140 $\pm$ 48 | 98 $\pm$ 48 |
| *S. natans* I | 163 $\pm$ 39 | 139 $\pm$ 39 | 192 $\pm$ 39 | 104 $\pm$ 39 |
| *S. natans* VIII | 58 $\pm$ 27 | 105 $\pm$ 27 | 69 $\pm$ 27 | 67 $\pm$ 27 |
| **Increase in internodes (number of new nodes $\cdot$ 10 d$^{-1}$)** | | | | |
| *S. fluitans* III | 3.7 $\pm$ 0.3 | 4.1 $\pm$ 0.4 | 4.5 $\pm$ 0.3 | 3.6 $\pm$ 0.4 |
| *S. natans* I | 1.3 $\pm$ 0.4 | 2.0 $\pm$ 0.6 | 2.4 $\pm$ 0.6 | 1.2 $\pm$ 0.4 |
| *S. natans* VIII | 2.5 [ab] $\pm$ 0.3 | 3.6 [a] $\pm$ 0.4 | 3.4 [ab] $\pm$ 0.3 | 2.1 [b] $\pm$ 0.4 |

In the first experimental series (*S. fluitans* III vs. *S. natans* VIII), the genotype $\times$ temperature interaction was significant (Figure 2; Table S2), suggesting that both genotypes grew differently at different temperatures. Unlike *S. natans* VIII, which maintained consistent growth rates over the temperature range, *S. fluitans* III peaked at 28 °C and then dropped at 31 °C. Nonetheless, *S. natans* VIII grew more slowly at temperatures above 28 °C than *S. fluitans* III at 28 °C. In the second experimental series, where *S. fluitans* III was compared to *S. natans* I, neither the main effect of the genotype nor the temperature nor the interaction of genotype $\times$ temperature were statistically significant (Figure 2; Table S2). In the third experimental series, *S. natans* I grew faster than *S. natans* VIII (Figure 2; Table S2). Neither the temperature nor the interaction, however, were significant (Table S2).

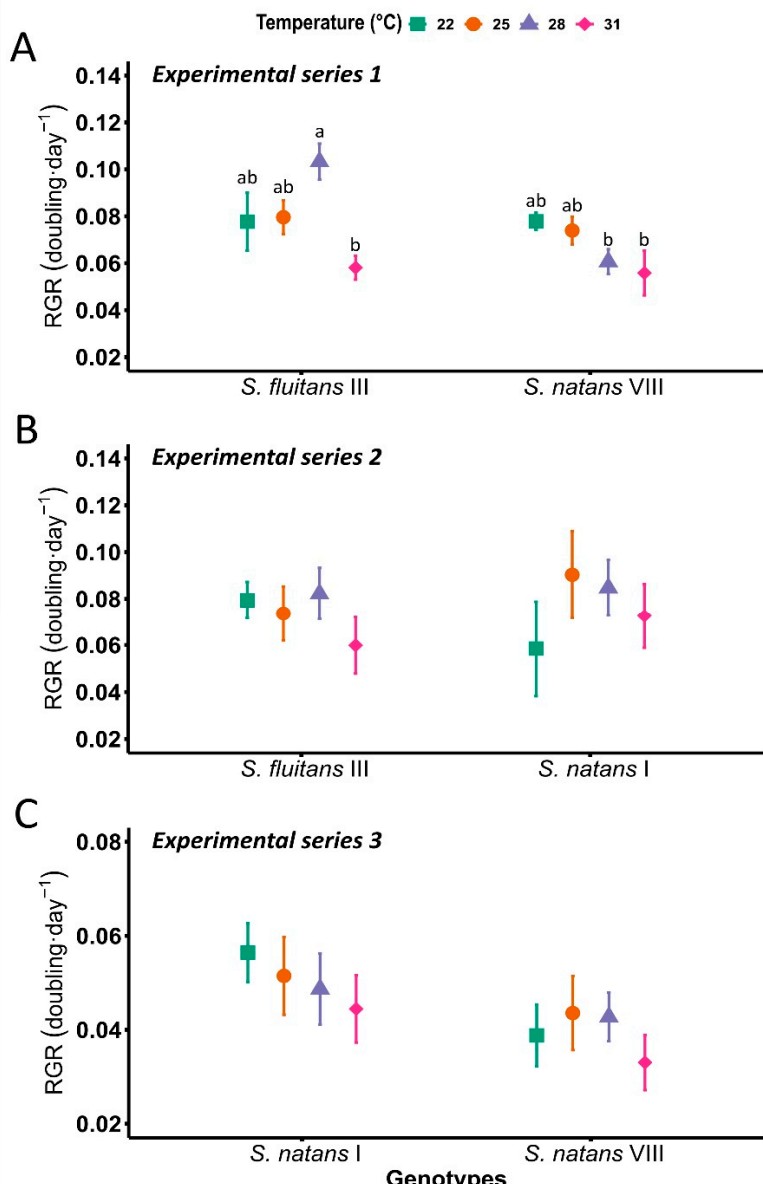

**Figure 2.** The relative growth rate (RGR, doubling d$^{-1}$; mean $\pm$ SE) of the holopelagic *Sargassum* genotypes in the ex situ experiments at different temperatures. (**A**) *S. fluitans* III vs. *S. natans* VIII, (**B**) *S. fluitans* III vs. *S. natans* I, and (**C**) *S. natans* I vs. *S. natans* VIII. Different letters indicate significant differences (post-hoc Tukey test, *p* = 0.05); two-way ANOVA where *p* < 0.05.

## 4. Discussion

The three genotypes of sargasso, responsible for the recurrent tropical Atlantic blooms, have different growth rates, as was also established in other works [14–17]. For the range of temperatures tested, *S. fluitans* III showed the highest rates of growth (Table 1). The growth rates determined in this study for the three genotypes of sargasso were the net growth rates, as the tissue loss during the trials was negligible. Only the youngest apical sections were used in the five-day trials, and the loss from the shedding or decay, was minimal (Table 1). The three sargasso genotypes had high growth rates over the entire temperature range in the Great Atlantic Sargassum Belt, albeit with some variation, which supports the statement by Skliris and collaborators [2] that "warming is not the driving the largest sargassum growth peaks over the last decade". Instead, nutrient supply (i.e., N and P), rather than temperature, were proposed as the principal drivers of the sargasso bloom [2,15]. However,

this does not imply that the temperature does not influence the phenology of sargasso at all, as it does for benthic *Sargassum* species [8–13] (pp. 9–14).

As hypothesized, the growth rates were generally lower at the highest temperature (31 °C) but not always (Figure 2). The three genotypes of sargasso did not have the same response to different temperatures. At 28 °C, *S. fluitans* III grew fastest (RGR 0.095 doubling $\cdot$ d$^{-1}$), whereas *S. natans* I did so at 25 °C (RGR 0.067 doubling $\cdot$ d$^{-1}$), and *S. natans* VIII at 22–25 °C (RGR 0.058–0.059 doubling $\cdot$ d$^{-1}$). *S. natans* VIII was the dominant genotype during the first blooms in 2011 [3], and a reduced tolerance to higher temperatures ($\geq$28 °C) was already shown for this genotype in an earlier work [17]. This supports the hypothesis proposed by Wang et al. [1] that the higher-than-usual seawater temperatures inhibited the formation of a sufficiently large seed population (of *S. natans* VIII) in 2010, despite all other conditions being favorable for blooming. Moreover, no bloom was observed in 2013, probably due to the lack of a vigorous populations, caused by higher-than-usual temperatures and a lack of nutrients [1,2]. The differential response to the temperature among the genotypes may also explain the reported gradual shift towards the predominance of *S. fluitans* III over time in the Great Atlantic Sargassum Belt, as this genotype grew generally faster than the other two genotypes, especially in the higher range of temperatures that prevail in the tropical Atlantic for a large part of the sargasso influx season. In the cooler Sargasso Sea, *S. natans* I typically dominated [3,6], although this may have changed recently, with the increasing influx of sargasso from the Great Atlantic Sargassum Belt over recent years [18].

The data analysis showed that, at the same temperatures, the growth rates of *S. natans* VIII were almost 50% lower in the third experimental series (autumn) than in the first series (spring; Figure 2). Examining the data by season, temperature, and genotype revealed this seasonal conditioning of growth applied to all genotypes (Figure 3; Table S3). This a posteriori analysis found that, in general, the growth rates (at the same temperature) were higher for the specimens collected in the spring and summer and lower for those collected in autumn (Figure 3; Table S3), at the end of the seasonal sargasso bloom cycle [1,2,6]. Since such seasonal conditioning was not part of the original research hypothesis, we did not obtain data for all the seasons and all the genotypes. Sand filtered seawater from the Puerto Morelos reef lagoon was used for the experiments, and the nutrient concentrations in this reef lagoon did not vary much with season during the study period [Van Tussenbroek et al. in prep.], excluding ambient nutrition as a cause for these differences. It is possible that the internal life cycle processes induced a reduction in growth rates later in the growing season; such changes are often induced by reproductive cycles, but sexual reproductive structures have not been found for any sargasso genotype to date [21]. Of special interest could be the study of temperature–nutrient interactions. On one hand, it has been known for decades that certain algal species, especially brown algae, including holopelagic *Sargassum* spp. can store nutrients in the vacuoles, which are used later for when other conditions (such as light and temperature) are suitable for vigorous growth [22–24]. On the other hand, tolerance of benthic *S. horneri* to elevated temperature was lower at high nitrogen availability [12], which may also occur for sargasso. Thus, even though the ambient nutrient availability in the culture systems was more or less constant, the internal nutrient reserves might have played a role in the growth rate differences with the season. The elemental concentrations in the sargasso tissues (including macro- and micronutrients) were highly variable, probably depending on the trajectory of the pelagic masses [25]. Therefore, it is recommended to determine the elemental composition (carbon, nitrogen, phosphorus, and possibly selected micronutrients) in future studies, whether nutrient analysis is the principal focus of the study or not, as the nutritional state of the sargasso possibly affects the outcome of any study on the physiology of these algae.

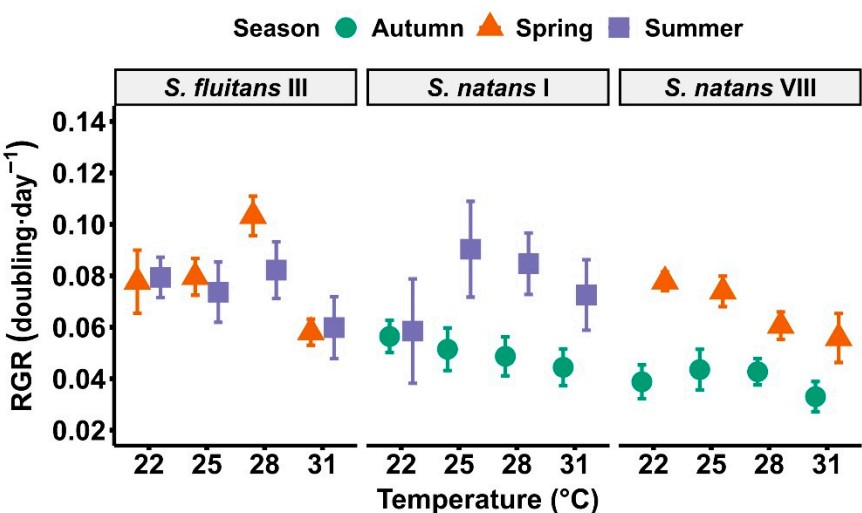

**Figure 3.** Relative growth rate (RGR, doubling $d^{-1}$; mean $\pm$ SE) of *S. fluitans* III, *S. natans* I, and *S. natans* VIII in ex situ experiments at different temperatures and in different seasons.

The study of the drivers or limiting parameters, such as the temperature, nutrient supply, salinity, irradiance, and biotic interactions on the growth of the sargasso is essential for understanding the bloom in the tropical North Atlantic [7]. Such knowledge would also help in the construction of more accurate (predictive) models (e.g., [2,18,26,27]), which in turn are necessary in planning the mitigation of the impacts of the bloom. It would also help in the processing of sargasso, as the high variability of the influx in time is a major challenge for the commercialization of collected sargasso [28–30]. Understanding the physiology of sargasso may even allow its cultivation at times of low influx to guarantee the supply of high-quality fresh seaweeds required for some products.

## 5. Conclusions

Sargasso showed high growth rates throughout the tested temperature range, and the different genotypes had distinct responses to temperature. *Sargassum fluitans* III generally presented higher rates than the other two genotypes, and all genotypes usually had decreased growth rates at 31 °C. Aside from temperature, many other factors could affect the growth of sargasso, and other relevant drivers, such as the nutrient reserves and the availability, salinity, light availability, their interactions, and possibly also inter-and intraspecific interactions, need to be considered. Studying the ecophysiology of sargasso has been challenging; we only obtained consistent results by maintaining the algae in motion [17]. The finding in this study that the response of the algae is variable under similar culture conditions depending on when they were collected poses yet another challenge. Even though at this moment we ignore whether these variable responses are due to their arrival history (e.g., having stored nutrients) or internal lifecycle mechanisms, this finding indicates that caution is needed comparing the growth rates of sargasso among sites and times. The study of the growth rates of sargasso under semi-controlled conditions, improves our understanding of the dynamics of the blooms of sargasso in the tropical North Atlantic.

**Supplementary Materials:** The following supporting information can be downloaded at: https:// www.mdpi.com/article/10.3390/phycology3010009/s1, Table S1: Details of the experimental series evaluating the growth of holopelagic Sargassum genotypes (sargasso) at different temperatures; Table S2: Two way ANOVA testing for differences in relative growth rates of sargasso by genotype and temperature. Bold values indicate statistical significance at $p < 0.05$; Table S3: Two-way ANOVA testing for differences in relative growth rate of sargasso by season and temperature. Bold values indicate statistical significance at $p < 0.05$.

**Author Contributions:** Conceptualization, E.M.-G. and B.I.v.T.; Methodology: E.M.-G., E.V.-M., E.R.S.-A., M.G.B.-S. and B.I.v.T.; formal analysis, E.M.-G., E.V.-M., E.R.S.-A., M.G.B.-S., R.S. and B.I.v.T.; investigation, E.M.-G., E.V.-M., E.R.S.-A., M.G.B.-S., R.S. and B.I.v.T.; resources, R.S. and B.I.v.T.; writing—original draft preparation, E.M.-G., R.S. and B.I.v.T.; writing—review and editing, E.M.-G., E.V.-M., E.R.S.-A., M.G.B.-S., R.S. and B.I.v.T.; project administration, E.M.-G., R.S. and B.I.v.T.; funding acquisition, R.S. and B.I.v.T. All authors have read and agreed to the published version of the manuscript.

**Funding:** This research was funded by DGAPA-UNAM, grant number PAPIIT IN203417, and Centro Mexicano de Innovación en Energía del Océano (CEMIE-Océano) funded by the CONACYT-SENER Sustentabilidad Energética project: FSE-2014-06-249795.

**Institutional Review Board Statement:** Not applicable.

**Informed Consent Statement:** Not applicable.

**Data Availability Statement:** The data presented in this study are available on request from the corresponding author.

**Acknowledgments:** We thank Elisa Vera Vázquez, Fernando Negrete-Soto, Edgar Escalante-Mancera, Miguel A. Gómez, Gustavo Villareal-Brito, Eduardo Ávila, and Laura Celis-Gutierrez for providing technical support or bibliographical resources.

**Conflicts of Interest:** The authors declare no conflict of interest. The funders had no role in the study design, data collection and interpretation, or the decision to submit the work for publication.

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
