# Peer review of "The Effect of Temperature on the Growth of Holopelagic Sargassum Species"

_phycology, doi:10.3390/phycology3010009_

Round 1

Reviewer 1 Report

Your paper deals with a pelagic Sargassum species in the northern tropical Atlantic, forming the "Great Atlantic Sargassum Belt", which is very important. However, I think that this paper should be re-written because the purpose of the study was not fully explained in the introduction.

In the Introduction you mentioned that ‘only one study has considered experiments relating temperature to the growth of sargasso from Sargasso Sea origin’. However, in the 2022 paper you published, the results of the analysis of the growth rate of the species have already been presented by a number of papers. Therefore, it is necessary to revise this part, and it is also necessary to explain the difference of this study again. After that, it is recommended to re-write the purpose of the study.

Materials and Methods is too simplistic to explain this experiment. In the growth experiment for the target species, since the growth characteristics may vary depending on the period of adaptation to the environment, an additional description of the acclimatization period and conditions is required. Also, it is difficult to understand that each experimental design was designed separately for each season until you see the results. The experimental period for each individual experiment also needs to be clearly presented.

In Results, Table 1 seems to express the results of each species as mean. However, since the number of replicates for each experiment period is different, it is necessary to explain how to reflect this and present the results. In addition, since the experimental period for each species is different, it is ambiguous to simply compare the growth of each species with this result.

A very important experiment was conducted for this paper, and it seems that good results were obtained. However, the premise and interpretation of the contents are lacking, and the most important justification for the experiment should be confirmed. In addition, there are many errors in each figure, table, and contents, and the completeness of the thesis is insufficient due to lack of unity (For example, In Table 1, the number of significant figures is inconsistent and the font size is different. / Line 131 – 0.45? / L154, 163, 171 – specie?). It is judged that it would be better to refer to the contents and rewrite it as a more complete paper.

In particular, the reference needs revision throughout [Inconsistent order (24 - 22, 23,) No specific number (1, 18), Inconsistent format, Inconsistent journal abbreviation, No italicization of species, Inconsistent capitalization and lowercase].

Author Response

We thank the reviewer for the thorough revision of our manuscript.

  1. The purpose of the study is not clear

“In the Introduction you mentioned that ‘only one study has considered experiments relatingtemperature to the growth of sargasso from Sargasso Sea origin’. However, in the 2022 paper you published, the results of the analysis of the growth rate of the species have already been presented by a number of papers. Therefore, it is necessary to revise this part, and it is also necessary to explain the difference of this study again. After that, it is recommended to re-write the purpose of the study.”

Answer: We agree with the referee that our statement “only one study had related temperature and growth of sargasso from Sargasso Sea origin” (i.e., that of Hanisak &Samuel 1987) may not have been entirely clear.  Other studies such as Lapointe et al. 2014, and 2021 (that included sargasso collected in different years from Sargasso Sea origin, as well as the GASP: Great Atl. Sargassum Belt), and that of some authors of this study that was recently published in Aquatic Botany 185(2023) 103614 (on two of the three genotypes from the GASB), also studied the growth of sargasso under different temperatures (and others that just mentioned ambient temperature when measuring growth), but these works never considered temperature as an explanatory variable for growth rates. They were not done in a proper series, and had different aims (e.g., Study the response to nutrients or testing a new system to cultures sargasso). Therefore, none of these studies could not and indeed did not draw conclusions concerning the response of these algae exposed to different temperatures along a range; leaving the study of Hanisak & Samuel in 1987 as the only other study analyzing response to temperature. Thus, we consider that the purpose of the study is justified; and our excuses that we did not phrase this sufficiently clear in the previous version.

We have changed our justification in the introduction

“However, to our knowledge, only one study has systematically related temperature to growth rates in experimental series [14]. But the sargasso in this study was from Sargasso Sea origin, and the employed resolution of temperatures, at increments of 6 °C from 18 °C to 30 °C, makes it difficult to determine the optimal temperature for growth. Others have studied the growth rates of sargasso for various purposes at different ambient temperatures [15-17], but their designs did not allow for drawing conclusions concerning the growth response of these algae to temperature.”

  1. Materials and Methods are too simplistic

“Materials and Methods is too simplistic to explain this experiment. In the growth experiment for the target species, since the growth characteristics may vary depending on the period of adaptation to the environment, an additional description of the acclimatization period and conditions is required. Also, it is difficult to understand that each experimental design was designed separately for each season until you see the results. The experimental period foreach individual experiment also needs to be clearly presented.”

Answer:

Thank you for pointing out that we omitted mentioning acclimation period and conditions: Algae collected at sea were acclimatized in the ex-situ culture system during one full day before the experiments, at prevailing ambient temperatures at the times of collection, which varied between 26.9 and 30.1 °C. This is now included in the text.

All trials lasted five days (first sentence in the section experimental design). In the discussion we clarified that we did not intended to include the factor “season” in the design, as we never expected that the time of the year would play a role in our experiments.  The experimental series were spread out over the “sargasso season” for logistical reasons, whenever we could collect fresh sargasso from the field and had access to the installations due to the COVID pandemic. A table in the supplement (Table S1) indicates now the period of each individual experiment.

We hope to have clarified the doubts in the design.

  1. Table 1

“In Results, Table 1 seems to express the results of each species as mean. However, sincethe number of replicates for each experiment period is different, it is necessary to explain howto reflect this and present the results. In addition, since the experimental period for eachspecies is different, it is ambiguous to simply compare the growth of each species with thisresult.”

Answer: We consider that presenting the average values across the temperature range of each sargasso genotype would provide useful general information on their general growth capacity. Although the design is not entirely balanced over the seasons (see above), the overall number of replicates per genotype per temperature are not very different (Table S1), allowing, in our opinion, for general comparisons of mean values; variations among genotypes and temperatures are also explained in the manuscript.

  1. Overall comment

“A very important experiment was conducted for this paper, and it seems that good results were obtained. However, the premise and interpretation of the contents are lacking, and the most important justification for the experiment should be confirmed. In addition, there are many errors in each figure, table, and contents, and the completeness of the thesis is insufficient due to lack of unity (For example, In Table 1, the number of significant figures is inconsistent and the font size is different. / Line 131 – 0.45? / L154, 163, 171 – specie?). It is judged that it would be better to refer to the contents and rewrite it as a more complete paper.

Answer: thank you for highlighting the relevance of this work, and we hope we have already explained the justification of this work our answer to point 1.

We have corrected the text: and instead of species we now use genotype: it is confusing that there are two (genetically different) morphotypes of S. natans, and we consider that indicating the species/morphotypes as “genotypes” may be most adequate. The figures are now edited and font sizes are the same.

  1. References

“In particular, the reference needs revision throughout [Inconsistent order (24 - 22, 23,) Nospecific number (1, 18), Inconsistent format, Inconsistent journal abbreviation, No italicizationof species, Inconsistent capitalization and lowercase].”

Answer: Thank you; the references have been revised and corrected

Reviewer 2 Report

There is no ref number 1, please add it to the document

Line 51, please upgrade the phrase "However, to our knowledge"

Line 76, since this is the first time UNAM appears in the document, please add the full name of the University

Figure 2. Please re-size all the figures. fig2c looks weird, and the same for fig3c and fig4c. Also, please revise fig 2b and 3b, and 4b which must be made again.

Figure 5 is too dense, with too much information, please try to break it in figure+table

There are no conclusions, please add this last part

Author Response

There is no ref number 1, please add it to the document

Answer: Reference no 1 is eliminated, and numbering is changed accordingly

Line 51, please upgrade the phrase "However, to our knowledge"

Answer:  the sentenced is changed as follows:

“However, to our knowledge, only one study has systematically related temperature to growth rates in experimental series [14]. But the sargasso in this study was from Sargasso Sea origin, and the resolution of temperatures employed, at increments of 6 °C from 18 °C to 30 °C, makes it difficult to determine the optimal temperature for growth”

Line 76, since this is the first time UNAM appears in the document, please add the full nameof the University

Answer: included

Figure 2. Please re-size all the figures. fig2c looks weird, and the same for fig3c and fig4c.Also, please revise fig 2b and 3b, and 4b which must be made again.

Answer: Thank you; the figures are now edited, and combined in one figure, and details of the ANOVA are mentioned in table S2

Figure 5 is too dense, with too much information, please try to break it in figure+table

Answer: We agree; the statistical data are now mentioned in table S3

There are no conclusions, please add this last part

Answer: Although we understand that conclusions are optional, we agree that the manuscript would benefit from addition of a concluding remark, and we have done so.

Reviewer 3 Report

 The occurrence of algae and algal blooms in marine, estuarine and freshwater ecosystems have increased dramatically worldwide due to increased anthropogenic introduction of nutrient-rich wastes into water bodies, effects of climate change and the continuous adaptation of algae to aquatic environments. The biology and ecology of macroalgae has been a subject of research, but there is still inadequate information on the factors favoring the proliferation of individual algal species in aquatic environments.

The report by Magaña-Gallegos et al. examines the influence of temperature as a contributory factor to the proliferation of Sargassum species (S. fluitans III, S. natans I and S. natans VIII) in the tropical North Atlantic. The study reports some interesting results (though examining a single factor), which could help in regulating the population of these algae in the region. I have the following comments on the submission.

COMMENTS

1. Title

Of the over 350 recognized species in the Sargassum genus, S. fluitans and S. natans are uniquely holopelagic. You may wish to therefore revise the current title to: Effects of temperature on the growth of holopelagic Sargassum species.

2. Abstract

L24-25: The sentence in these lines should be omitted, and replaced with the implications of the results obtained. For example, based on the results of the study, what do the authors now recommend? Temperature is just one factor that affects the growth of (macro)algae, and therefore other factors (e.g., pH, sunlight, nutrient loads and buoyancy) needs to be investigated too.

2. Keywords

L26:  bloom >> algal bloom

3. Methods

L112: was 200 to 300 mol m−2 s−1 >> (200 to 300 mol m−2 s−1); L118: growth >> grow

L154, 163, 171, 240: Please correct ‘‘specie’’ to species throughout the manuscript.

4. Discussion

Discussion should be improved, by bringing in comparison with previous reports on other Sargassum species; e.g.

Wu et al. 2022. https://doi.org/10.3390/jmse10111692

Zhou et al. 2017. https://doi.org/10.1007/s10811-017-1282-4

5. References

Some references are incompletely/incorrectly captured. For example,

L291-292: volume no. is missing; the publication year is incorrect and indicated twice.

L315-316: Ref. 17 has no DOI or page ranges nor volume. If it is still in press/just accepted, it should be indicated so.

L345: [30], the article no. is not correct.

Author Response

Thank you  for describing the importance of this work

  1. Title

“Of the over 350 recognized species in the Sargassum genus, S. fluitans and S. natans areuniquely holopelagic. You may wish to therefore revise the current title to: Effects of temperature on the growth of holopelagic Sargassum species”.

Answer: We agree, and this has been changed in he title and throughout the text

  1. Abstract

“L24-25: The sentence in these lines should be omitted, and replaced with the implications of the results obtained. For example, based on the results of the study, what do the authors now recommend? Temperature is just one factor that affects the growth of (macro)algae, and therefore other factors (e.g., pH, sunlight, nutrient loads and buoyancy) needs to be investigated too.”

Answer: We agree, the message in our concluding sentence of the abstract was limited; we have now changed this to:

“Understanding the role of temperature in the growth of holopelagic Sargassum species, amongst other factors influencing the physiology (such as nutrients, salinity tolerance and light, including their interactions), could help to understand the recent blooms in the tropical North Atlantic.”

Of course, there many other relevant aspects, such as intra- and interspecific interactions, studying the mortality; but mentioning these would be out of the scope of this work.

  1. Methods

Answer: We thank the reviewer for pointing out several typing errors, which have all been corrected

  1. Discussion

“Discussion should be improved, by bringing in comparison with previous reports on other Sargassum species; e.g. Bui et al. 2018. https://thescipub.com/pdf/ajassp.2018.186.197.pdf, Wu et al. 2022. https://doi.org/10.3390/jmse10111692 , Zhou et al. 2017. https://doi.org/10.1007/s10811-017-1282-4”

Answer: Thank you for pointing out these studies, although the focus of these studies (i.e. cultivating Sargassum species for commercial purposes) than this work (i.e. understanding the physiology of holopelagic species), their finding are relevant to our study, and we have now included the first two references (replacing previously mentioned more general and less relevant references).  The suggested work by Zhou et al. 2017 is not included, because this studies the tolerance of tropical species far below the ambient temperature and light conditions of the species, possibly to discern minimal conditions for purposes; and although the high tolerance they found is interesting, and relevant in comparing tolerance of holopelagic species (not commonly exposed to varying salinities in the tropical Atlantic) and benthic species (some may be exposed to fluctuating salinity); this is outside the scope of this study.

We are aware that many more studies on the growth and physiology of benthic Sargassum species exist; but we can´t mention them all, and sought out those that aided in enriching the introduction and discussion.

  1. References

Answer: We thank the reviewer for pointing out several typing errors, which have all been corrected

Round 2

Reviewer 1 Report

The revised manuscript was well structured, and the reviewer's opinions were well reflected.

Please check a few things.

Method: It is recommended to add contents related to Table S3 to 2.4 Statistical analysis.

Table 1. It is necessary to check the font size (number in ‘Time to double weight’), and show whether it is mean SE.

Conclusion: I recommended to add summary of the main results of your paper.

Reference: 4. Check journal name abbreviation.

Author Response

The revised manuscript was well structured, and the reviewer's opinions were well reflected.

Answer: Thank you for your suggestions for improving the manuscript.

Method: It is recommended to add contents related to Table S3 to 2.4 Statistical analysis.

Answer: This statistical analysis was added to section 2.4.

A factorial ANOVA analysis was also performed for each genotype, with the experimental series (realized in different seasons) and temperature as factors, as well as their interactions; this study was performed after we, observed, against our expectations, that the growth rates of S. natans VIII differed between experimental series.”

Table 1. It is necessary to check the font size (number in ‘Time to double weight’), and show whether it is mean SE.

Answer: Font size was corrected and now it is displayed that all data are mean ± SE.

Conclusion: I recommended to add summary of the main results of your paper.

Answer: Conclusion was improved with a summary of the main results.

Reference: 4. Check journal name abbreviation.

Answer: Thank you, the full journal name is now mentioned

Note: In addition, we have made few minor corrections to the text.